# Advancing Colorectal Cancer Diagnostics from Barium Enema to AI-Assisted Colonoscopy

**DOI:** 10.3390/diagnostics15080974

**Published:** 2025-04-11

**Authors:** Dumitru-Dragos Chitca, Valentin Popescu, Anca Dumitrescu, Cristian Botezatu, Bogdan Mastalier

**Affiliations:** 1General Surgery Clinic, Colentina Clinical Hospital, 020125 Bucharest, Romania; popescu.vali.umf@gmail.com (V.P.); cristian.botezatu@umfcd.ro (C.B.); bogdanmastalier@yahoo.com (B.M.); 2General Surgery Clinic, Carol Davila University of Medicine and Pharmacy, 050474 Bucharest, Romania; 3Family Medicine, Vitan Polyclinic, 031087 Bucharest, Romania; ancadumi@gmail.com

**Keywords:** colorectal cancer, diagnostic tools, ai-assisted colonoscopy, liquid biopsy, survival, genetic test

## Abstract

Colorectal cancer (CRC) remains a major global health burden, necessitating continuous advancements in diagnostic methodologies. Traditional screening techniques, including barium enema and fecal occult blood tests, have been progressively replaced by more precise modalities, such as colonoscopy, liquid biopsy, and artificial intelligence (AI)-assisted imaging. **Objective**: This review explores the evolution of CRC diagnostic tools, from conventional imaging methods to cutting-edge AI-driven approaches, emphasizing their clinical utility, cost-effectiveness, and integration into multidisciplinary healthcare settings. **Methods**: A comprehensive literature search was conducted using the PubMed, Medline, and Scopus databases, selecting studies that evaluate various CRC diagnostic tools, including endoscopic advancements, liquid biopsy applications, and AI-assisted imaging techniques. Key inclusion criteria include studies on diagnostic accuracy, sensitivity, specificity, clinical outcomes, and economic feasibility. **Results**: AI-assisted colonoscopy has demonstrated superior adenoma detection rates (ADR), reduced interobserver variability, and enhanced real-time lesion classification, offering a cost-effective alternative to liquid biopsy, particularly in high-volume healthcare institutions. While liquid biopsy provides a non-invasive means of molecular profiling, it remains cost-intensive and requires frequent testing, making it more suitable for post-treatment surveillance and high-risk patient monitoring. **Conclusions**: The future of CRC diagnostics lies in a hybrid model, leveraging AI-assisted endoscopic precision with molecular insights from liquid biopsy. This integration is expected to revolutionize early detection, risk stratification, and personalized treatment approaches, ultimately improving patient outcomes and healthcare efficiency.

## 1. Introduction

Colorectal cancer (CRC), including colon cancer and rectal cancer, represents a significant public health challenge globally, not only due to its high morbidity and mortality rates but also because of its substantial economic burden. As the third most common malignancy worldwide, CRC is responsible for a growing proportion of healthcare expenditures, particularly in high-income countries where screening programs are widely implemented. The costs associated with CRC include direct medical expenses, such as diagnostic procedures, treatment interventions, and long-term management, as well as indirect costs, including loss of productivity and financial stress on patients and caregivers [1,2,3,4].

The economic impact of CRC screening and diagnosis varies based on factors, such as the type of screening method used, healthcare system efficiency, and national policies. Traditional screening techniques, such as fecal occult blood tests (FOBT) and colonoscopy, have proven to be cost-effective strategies for early detection, even if adherence to screening programs remains suboptimal, especially in lower-income groups [5].

Liquid biopsy and AI-assisted colonoscopy have emerged as promising diagnostic advancements. Their high upfront costs and the need for further validation pose challenges at this stage of implementation [6].

Early detection through cost-effective screening significantly reduces overall healthcare expenses by identifying CRC at an early, treatable stage, thereby lowering the need for expensive, late-stage treatments [7]. Financial barriers, limited accessibility, and disparities in healthcare infrastructure continue to hinder optimal CRC screening and diagnosis [8].

CRC develops through two well-established molecular pathways: the suppressor (chromosomal instability) pathway and the mutator (microsatellite instability) pathway. The suppressor pathway, which accounts for approximately 80% of sporadic colorectal cancers, follows a stepwise progression from benign adenomas to malignant tumors. In contrast, the mutator pathway, seen in about 20% of sporadic CRCs and 80% of hereditary cases, is characterized by mutations in genes such as KRAS (40%), APC (60%), TP53 (70%), DCC (70%), and BAX (50%) [9,10,11].

The disease typically originates as a small polyp within the intestinal lining, which may progress into malignancy based on factors such as histological type and lesion size. About 60% of cases involve single adenomas, while 40% present as multiple adenomas. If left untreated, approximately 24% of patients with polyps eventually develop CRC [12].

## 2. Materials and Methods

This review was conducted to evaluate advancements in CRC diagnostic tools, with a particular focus on AI-assisted colonoscopy, liquid biopsy, and traditional screening methods. The study followed the PRISMA (Preferred Reporting Items for Systematic Reviews and Meta-Analyses) guidelines to ensure a structured and unbiased selection of relevant scientific literature.

A comprehensive literature search was performed using PubMed, Medline, Scopus, and Web of Science. The search strategy was developed based on predefined keywords and Boolean operators, ensuring coverage of essential CRC diagnostic methodologies. The keywords included “colorectal cancer diagnostics”, “AI-assisted colonoscopy”, “liquid biopsy CRC”, “adenoma detection rate (ADR)”, “tumor markers”, and “screening tools for colorectal cancer”.

Structured search queries were designed to enhance retrieval accuracy and ensure comprehensive literature coverage. The results from these queries were systematically compared with manually entered search criteria to ensure comprehensive coverage of relevant studies.

Studies were selected based on specific inclusion and exclusion criteria to ensure relevance and quality. Eligible studies were those published between 2015 and 2025, focusing on recent advancements in CRC diagnostics. Only peer-reviewed articles written in English were considered, emphasizing AI-assisted colonoscopy, liquid biopsy, and traditional CRC screening methods. The review included clinical trials, systematic reviews, meta-analyses, and cohort studies relevant to CRC diagnostics.

Studies were excluded if they were not published in English, lacked direct relevance to CRC diagnostics, or did not contribute meaningful clinical insights. Research with small sample sizes or insufficient methodological rigor was also excluded.

## 3. Results

### 3.1. Epidemiology

According to the latest GLOBOCAN 2022 report, CRC is the third most commonly diagnosed cancer worldwide, with an estimated 1,926,425 new cases in 2022, accounting for 9.6% of all cancer cases. In males, CRC ranks third, with 1,069,446 new cases, making up 10.4% of all male cancer diagnoses. Among females, it is the third most common cancer, with 856,979 new cases, representing 8.9% of all female cancer cases. In terms of mortality, CRC remains a significant burden, ranking as the second leading cause of cancer-related deaths globally, with an estimated 904,019 deaths in 2022. The mortality burden is particularly higher in low- and middle-income regions, where access to early screening and advanced treatment remains limited [2].

In 2022, the estimated global incidence had the highest rates observed in developed nations [13]. The incidence rate in high-income countries remains significantly higher compared to low- and middle-income regions, where access to screening and early detection programs is often limited [14].

Despite advancements in CRC screening and early detection, global incidence trends have shown divergent patterns. In many high-income countries, incidence rates have begun to stabilize or decline, largely due to widespread adoption of colonoscopy screening and removal of precancerous lesions [15]. Concerning trends have emerged in low- and middle-income countries, where CRC incidence is steadily increasing due to shifts in lifestyle factors, dietary habits, and aging populations [16].

Early-onset CRC (diagnosed before age 50) is rising, particularly in developed nations, though an overall decline is present in older populations [17].

According to the INEGI, cancer accounted for 74,685 deaths in Mexico in 2010, representing 13% of total mortality, with CRC responsible for 5.4% of these cases. CRC can arise sporadically due to non-hereditary mutations, transcriptional silencing of tumor suppressor genes, and alterations in genes regulating the cell cycle, DNA repair, and apoptosis. It can also have a genetic origin, involving mutations in tumor suppressor genes, such as APC, DCC, BRAF, PIK3CA, AKT, and TP53, or the activation of oncogenes such as K-RAS and CTNNB1 [18] (Table 1).

CRC arises due to chromosomal abnormalities, genetic mutations, and epigenetic modifications that impact key cellular processes such as proliferation, differentiation, apoptosis, and angiogenesis [19]. Among molecular markers, the epidermal growth factor receptor (EGFR) is widely recognized for its role in predicting treatment response. One of the most frequently mutated oncogenes in sporadic CRC is K-RAS, which is located on chromosome 12. Approximately 95% of K-RAS mutations occur in codons 12 and 13, while the remaining 5% involve codons 61, 146, and 154. These mutations are more commonly observed in patients with metastatic disease. There is a known correlation between mutations at codon 12 and mucinous-type colorectal cancer, whereas codon 13 mutations are typically associated with non-mucinous CRC, a more aggressive subtype with higher metastatic potential [7,8,11,20,21].

K-RAS mutations, particularly those in codon 13, are linked to poor prognosis and are considered biomarkers for predicting therapy resistance via the EGFR pathway. The genetic development of CRC often starts with an inactivating mutation of the APC gene, which is responsible for familial adenomatous polyposis (FAP). This APC mutation is present in approximately 85% of non-hereditary CRC cases and plays a critical role in initiating the adenoma–carcinoma sequence.

Some adenocarcinomas develop following the mutational activation of β-catenin (CTNNB1), regulated by APC, or through an alternative pathway involving the inactivation of tumor suppressor genes responsible for DNA repair. These genes, known as mismatch repair (MMR) genes, include MSH2 (mutS homolog 2), MLH1 (mutL homolog 1), and PMS2 (postmeiotic segregation increased 2), and are implicated in both hereditary syndromes and sporadic colorectal cancer. Mutations in colorectal carcinogenesis typically follow a sequential pattern, with initial alterations affecting APC, followed by mutations in RAS [22].

Mutations in the TP53 tumor suppressor gene are detected in approximately 50% of CRC cases and tend to occur during the later stages of tumor progression. In terms of somatic genetic alterations, the WNT/β-catenin signaling pathway, which is essential for cell proliferation and tissue homeostasis, is frequently disrupted in CRC. Dysregulation of this pathway is linked not only to cancer, but also to other conditions, such as congenital abnormalities and osteoporosis. Notably, WNT/β-catenin pathway mutations are present in up to 95% of CRC patients [9,23,24,25]. The rs59336 polymorphism, found in the TBX3 gene (a key component of the WNT/β-catenin pathway), is associated with increased susceptibility to CRC. This genetic variant may contribute to alterations in transcriptional regulation and tumor initiation in individuals predisposed to CRC [26].

Genetic alterations in the SMAD7 gene have been implicated in the progression of CRC. Specific variants of SMAD7, particularly rs44939827, rs12953717, and rs4464248, located on chromosome 8q21, have been identified as risk factors for CRC. These polymorphisms are thought to modulate TGF-β signaling, which plays a crucial role in cell differentiation, proliferation, and apoptosis. Dysregulation in this pathway can contribute to tumor initiation and progression [27,28].

Metastases protective genes, such as KISS1 have been identified, which connect to the KISS receptor. This is essential for metastasis suppression, and the survival rate significantly improves to high levels of KISS1 and KISS1R [29,30].

### 3.2. Risk Factors

Among the risk factors associated with CRC, 30–40% can be attributed to hereditary factors. Family history plays a significant role in CRC risk, particularly in individuals with hereditary conditions, such as familial adenomatous polyposis and Lynch syndrome (hereditary non-polyposis colorectal cancer, HNPCC) [31]. These syndromes are linked to mutations in genes involved in the MMR pathway, specifically MLH1, MSH2, MSH6, and PMS2. Mutations in MLH1 and MSH2 account for approximately 90% of the mutations identified in families with hereditary CRC, both with and without polyposis.

In contrast, germline mutations in APC, MTHYH, SMAD4, BMPR1A, and STK11 contribute to less than 5% of all CRC cases [32]. It is estimated that these hereditary genetic syndromes represent approximately 10% of all CRC cases. In about 25% of cases, a family history of CRC increases the risk of developing the disease, even in the absence of an identified genetic syndrome [33].

Individuals with a history of inflammatory bowel diseases (IBD), such as ulcerative colitis or Crohn’s disease, along with those diagnosed with personal or familial polyposis syndromes, have a 30–50% increased risk of developing CRC. Additionally, prior diagnoses of colon, rectal, ovarian, endometrial, or breast cancer, as well as diabetes mellitus, have been linked to a higher likelihood of CRC development. However, it is important to note that approximately 75% of colorectal malignancies arise in individuals without any of these predisposing conditions.

The association between hyperplastic polyposis and CRC remains debated. Adenomatous polyps are frequently found in individuals over the age of 50, but not all polyps progress to malignancy. The histological characteristics and size of a polyp play crucial roles in determining its potential for malignant transformation. A higher risk to developing malignant characteristics in hyperplastic polyps include polyp size greater than 10 mm, the presence of dysplasia, and localization in the right colon. The presence of an adenomatous component within a hyperplastic polyp (mixed hyperplastic–adenomatous polyp), more than 20 hyperplastic polyps in the colon, and a family history of colorectal cancer are also associated with an increased risk of malignancy.

The progression of these polyps to malignancy typically follows a well-defined evolutionary timeline of 10–15 years, beginning as mild dysplasia and advancing to moderate or severe dysplasia, depending on the accumulation of genetic alterations. The pathways of tumor dissemination are primarily hematogenous or lymphatic, which influence the growth rate and disease progression. Rare cases of tumor implantation following surgical manipulation, particularly after laparoscopic colectomy, have also been reported [3].

### 3.3. Prophylactic Factors

The use of non-steroidal anti-inflammatory drugs (NSAIDs) has been shown to reduce the risk of CRC. The molecular basis of this protective effect involves the regulation of epidermal growth factor receptor (EGFR) overexpression, an early event in colorectal tumor genesis. EGFR is overexpressed in approximately 80% of CRC cases. Additionally, the overexpression of cyclooxygenase-2 (COX-2) activates the c-Jun-dependent transcription factor activator protein-1 (AP-1), which binds to the EGFR promoter. As a result, selective COX-2 inhibitors have been proposed as chemopreventive agents against CRC [34]. Dietary factors also play a crucial role in CRC prevention. While high dietary fiber intake is associated with a reduced risk of CRC and other malignancies, findings regarding the protective effects of fruit and vegetable consumption remain inconclusive. No significant association has been found between total fruit and vegetable intake and CRC risk when analyzed separately [35]. While variety in overall fruit and vegetable consumption does not appear to lower CRC risk, a high variety in fruit intake alone has been linked to a 41% increased risk for individuals consuming more than eight different fruits every two weeks [36].

### 3.4. Diagnostic Methods

For individuals at moderate risk and aged 50 or older, current colorectal cancer screening strategies include several options. One approach is the fecal occult blood test (FOBT), which utilizes high-sensitivity Guaiac-based or immunological testing (FIT) and is recommended annually. Another method involves sigmoidoscopy every five years, supplemented by FOBT every three years. Alternatively, a colonoscopy is advised once every ten years as a comprehensive screening measure [37]. Diagnostic methods play a crucial role in determining the overall survival of patients with CRC. FOBT can yield false-positive results (Table 2). These inaccuracies may arise due to high consumption of red meat, fruits, and vegetables containing peroxidase, which can mimic the presence of blood in stool.

On the other hand, individuals without gastrointestinal bleeding who consume high doses of vitamin C may experience false-negative results, further complicating diagnostic accuracy [4]. Advancements in molecular technology have led to the development of more precise diagnostic tools, such as polymerase chain reaction (PCR), which enables the identification of genetic biomarkers associated with CRC. PCR-based assays allow for the detection of specific mutations in genes involved in colorectal carcinogenesis, improving early detection and risk assessment.

One emerging biomarker for CRC is pancreatic duodenal homeobox-1 (PDX-1), a transcription factor involved in pancreatic development and cellular proliferation. Elevated PDX-1 expression has been observed in malignant tumors of the pancreas, breast, colon, prostate, and kidney, as well as in metastatic cancers. In contrast, low or undetectable levels of PDX-1 have been reported in primary tumors and normal colonic tissue, suggesting its potential role as a biomarker for CRC diagnosis and progression [38,39,40].

Among diagnostic methods, sigmoidoscopy continues to advance, improving its detection rates; 65–75% of adenomatous polyps and 40–65% of colorectal cancers are within the reach of this procedure. Approximately 50% of advanced adenomas (>1 cm) and proximal colon cancers remain undetectable using sigmoidoscopy alone [4].

Despite its role in screening, diagnosis, and therapeutic management, sigmoidoscopy has notable limitations. Compared to other CRC screening tests, it is associated with higher costs, increased procedural risks, and greater discomfort for patients, which may affect adherence to screening recommendations [41].

The overexpression of certain proteins and genetic factors can serve as prognostic markers in CRC. Among these, elevated mRNA and paxillin levels have been associated with histological tumor grade, tumor size, clinical TNM stage, and distant metastases. Studies indicate that patients with high paxillin expression have an unfavorable prognosis compared to those with low paxillin expression [42,43].

Additionally, pre-treatment hypoalbuminemia (<3.5 g/dL) has been identified as an independent predictor of lower post-resection survival, particularly in patients with stage II CRC. Given its prognostic significance, hypoalbuminemia may serve as a simple and effective biomarker for poor prognosis, available at the time of diagnosis, aiding in early risk stratification and treatment planning [44,45].

### 3.5. Survival

The five-year survival rate represents the proportion of patients who remain alive at least five years post-diagnosis, factoring in the cancer type, disease stage, and treatment approach. This metric is also influenced by genomic alterations in cancer cells and individual biological variability. In cases of CRC, the relative survival rates are estimated to be approximately 65% at five years and 58% at ten years [46]. A 2025 study by Mihanfar et al. reaffirms the prognostic significance of lactate dehydrogenase (LDH) levels in cancer progression and survival. Their findings indicate that elevated LDH levels are associated with poorer outcomes in CRC, as well as in other malignancies, such as prostate, lung, gastroesophageal, and gynecologic cancers. Increased LDH levels correlate with higher tumor burden, increased metabolic reprogramming in cancer cells, and enhanced tumor aggressiveness, making LDH a valuable biomarker for early risk stratification and treatment planning. Additionally, another study by Al-Abady et al. (2024) highlights the combined role of LDH and caspase-3 as potential biomarkers for mediating CRC therapy, further supporting LDH’s role as an indicator of treatment response and disease progression [47,48].

The survival of patients diagnosed with CRC varies depending on the cancer type and stage. Patients with stage IIIA or IIIB CRC have been reported to exhibit better survival rates than those diagnosed with stage IIB. Similarly, in cases of rectal cancer, certain stage III patients demonstrate higher survival rates compared to some individuals with stage II disease, highlighting the complexity of CRC prognosis and treatment response.

Patients with stage III CRC harboring KRAS mutations had worse survival outcomes compared to those with wild-type KRAS (KRAS-WT), indicating the oncogene’s impact on disease progression and prognosis.

The expression of KISS1 and its receptor, KISS1R, has been identified as a prognostic factor in CRC survival. Higher expression levels of KISS1/KISS1R correlate with improved survival rates, increasing from 44.3% and 39.3% to 73.7% and 67.9%, respectively. Patients with low KISS1 expression were more likely to develop distant metastases, suggesting that KISS1 could serve as a promising prognostic and therapeutic biomarker in CRC. These findings align with previous research by Zhu et al., which reported that KISS1 is associated with improved outcomes by inhibiting matrix metalloproteinase-9 (MMP-9) activity in colorectal liver metastases [49].

Survival rates in CRC vary significantly depending on the disease stage. The five-year survival rate is approximately 90% for localized CRC, 68% for regional disease with lymphatic involvement, and 10% for metastatic disease. This contrast underscores the importance of early detection, as CRC prognosis is more closely linked to tumor stage at diagnosis than to tumor size, unlike other malignancies where tumor dimensions play a more significant prognostic role.

The increasing CRC-related mortality rate can be attributed to rising life expectancy and age-associated genetic alterations, leading to a higher susceptibility to carcinogenic factors, cumulated with immune suppression and comorbidities in aging populations [35].

### 3.6. Liquid Biopsy

Liquid biopsy refers to the analysis of circulating tumor DNA (ctDNA), circulating tumor cells (CTCs), extracellular vehicles (EVs), and microRNAs (miRNAs) in bodily fluids, such as blood, plasma, or urine. This technique enables real-time monitoring of tumor dynamics and offers several advantages over conventional tissue biopsies, including early detection, minimal invasiveness, and the ability to track tumor evolution over time [50]. Several key biomarkers have been identified for CRC diagnosis and prognosis through liquid biopsy as seen in Table 3.

Beyond its role in diagnosis, liquid biopsy has emerged as a valuable tool in various stages of CRC management. One of its most significant applications is in early detection and screening, where it provides a non-invasive alternative to traditional diagnostic methods. This approach is particularly beneficial for high-risk populations who may not undergo routine colonoscopy. A recent study demonstrated that liquid biopsy achieved an 85–90% accuracy rate in detecting CRC among asymptomatic individuals, highlighting its potential as an effective screening modality [56].

In addition to early detection, liquid biopsy holds prognostic and predictive value by assessing ctDNA levels, which correlate with tumor burden and response to therapy. Elevated ctDNA levels have been linked to worse clinical outcomes, making them reliable biomarkers for disease progression. Furthermore, the presence of KRAS and BRAF mutations in ctDNA has been associated with resistance to anti-EGFR therapies, enabling clinicians to refine treatment strategies and improve personalized oncology approaches [57].

Another critical application of liquid biopsy is in the detection of minimal residual disease (MRD) following surgery or chemotherapy. By identifying micrometastatic disease that may not be visible on imaging, liquid biopsy allows for early intervention and adjustment of therapeutic regimens. Studies indicate that patients who test positive for ctDNA post-surgery face a significantly higher risk of relapse, necessitating closer surveillance and more aggressive adjuvant treatment plans [58].

Moreover, liquid biopsy has revolutionized the monitoring of treatment response, providing real-time insights into tumor dynamics without the need for invasive procedures. By tracking ctDNA fluctuations over time, clinicians can assess the efficacy of ongoing treatments and make timely modifications. A study on metastatic CRC patients found that serial ctDNA monitoring closely correlated with disease progression and response to immunotherapy, reinforcing its potential as a dynamic tool for personalized cancer care [59].

Liquid biopsy offers a versatile and minimally invasive approach to CRC management, spanning early detection, prognosis, treatment monitoring, and MRD assessment. As advancements in molecular diagnostics continue, liquid biopsy is poised to play an increasingly integral role in precision oncology, ultimately improving patient outcomes and treatment strategies (Figure 1).

Despite its promising potential, liquid biopsy encounters several technical and clinical challenges that must be addressed before it can fully replace conventional diagnostic methods. One of the primary concerns is sensitivity and specificity limitations, particularly in early-stage CRC. Differentiating ctDNA from normal circulating DNA remains a significant challenge, as the fraction of ctDNA in blood samples can be exceedingly low, making detection difficult and potentially leading to false-negative results.

Another major obstacle is the lack of standardization and regulatory approval across different ctDNA detection methods and assay platforms. Variability in analytical techniques, thresholds for ctDNA positivity, and assay sensitivity creates inconsistencies in test results, hindering widespread clinical adoption. Without uniform guidelines and regulatory frameworks, implementing liquid biopsy as a routine diagnostic tool remains a challenge.

Cost and accessibility further complicate the integration of liquid biopsy into standard cancer care. While this approach is less invasive than traditional tissue biopsy, the high cost of testing and limited availability in low-resource settings present barriers to global implementation. For many healthcare systems, particularly in developing countries, the financial burden of advanced molecular diagnostics limits their adoption in routine clinical practice.

Beyond these technical and economic challenges, large-scale validation through prospective clinical trials and real-world studies is essential to establish the clinical utility of liquid biopsy in CRC management. While early studies show promising results, comprehensive longitudinal research is required to assess its effectiveness in different patient populations, treatment settings, and disease stages.

Looking ahead, the integration of artificial intelligence (AI) and machine learning into liquid biopsy analysis offers a potential solution to many of these challenges. AI-driven algorithms could enhance mutation detection accuracy, improve risk stratification, and optimize ctDNA analysis, leading to more reliable and cost-effective applications. By refining diagnostic precision and overcoming current limitations, AI-enhanced liquid biopsy could transform cancer diagnostics, making early detection, treatment monitoring, and personalized therapy more accessible and effective for patients worldwide.

### 3.7. Ai-Assisted Imaging and Deep Learning in Colonoscopy and CRC Diagnostic

#### 3.7.1. Uses and Benefits

Colonoscopy is the gold standard for CRC screening. Its effectiveness is influenced by operator expertise, leading to variability in ADR and missed lesions, which contribute to interval cancers [60]. To address these limitations, AI has emerged as a transformative tool in colonoscopy, leveraging deep learning (DL) and machine learning (ML) to enhance real-time decision making and diagnostic precision [61].

AI-assisted imaging systems utilize large-scale annotated datasets of colonoscopic images to train deep neural networks, enabling automated polyp detection, classification, and quality assessment [62]. These models, particularly convolutional neural networks (CNNs) and transformer-based architectures, have demonstrated high accuracy in lesion recognition, significantly reducing miss rates and improving standardization in CRC screening [63,64].

AI-powered Computer-Aided Detection (CADe) systems assist endoscopists by identifying polyps in real time, improving ADR and reducing miss rates [64]. Deep learning models, particularly CNNs, have demonstrated superior performance in polyp segmentation and early lesion detection [61].

AI-based Computer-Aided Diagnosis (CADx) systems differentiate benign from malignant lesions using advanced imaging analytics. By analyzing polyp morphology, vascularization, and surface texture, deep learning models achieve diagnostic accuracies exceeding 90%, reducing unnecessary biopsies and polypectomies [60].

Lymph Node (LN) Detection uses AI to enhance the preoperative staging of CRC by identifying metastatic lymph nodes in radiological imaging. Machine learning algorithms applied to computed tomography (CT) and magnetic resonance imaging (MRI) scans improve the sensitivity and specificity of LN detection, aiding surgical planning and treatment stratification [63].

AI models can predict treatment response in CRC patients undergoing neoadjuvant therapy. By analyzing multimodal data—including histopathological images, genomic profiles, and radiologic scans—AI algorithms provide accurate assessments of complete pathological response [63]. These predictions help personalize treatment approaches and minimize overtreatment. AI can also facilitate long-term disease management by predicting CRC recurrence and patient prognosis. Machine learning models trained on clinical and pathological datasets identify high-risk patients, enabling early intervention strategies [65].

#### 3.7.2. Architecture Types

Various deep learning architectures, including CNNs, transformer-based models, and hybrid AI frameworks, have demonstrated distinct advantages in improving polyp detection and segmentation, thereby enhancing the overall accuracy and efficiency of CRC screening.

CNNs have emerged as the dominant deep learning architecture for CADe and CADx in colonoscopy. CNN-based AI models analyze high-resolution colonoscopic images to identify and segment polyps with a performance comparable to experienced endoscopists. CNN-based systems, such as DeepLabV3+ and U-Net, have demonstrated exceptional polyp segmentation capabilities, accurately distinguishing between neoplastic and non-neoplastic lesions [60] Studies show that CNN models achieve an ADR increase of up to 14%, significantly reducing the risk of interval cancers [62].

A deep-learning-based AI model developed for real-time polyp detection achieved an accuracy of over 90%, outperforming traditional methods reliant on clinician expertise [61].

Transformer-based architectures, originally designed for natural language processing (NLP), have recently been adapted for colonoscopy due to their superior ability to analyze sequential video frames and improve real-time polyp localization. Unlike CNNs, transformers capture long-range dependencies, making them particularly useful for continuous endoscopic video analysis. Transformer-based AI models, such as Swin-Transformer and Vision Transformer (ViT), have demonstrated enhanced feature extraction, reducing the rate of false-positive detections [66].

A comparative study found that transformer-based models outperformed CNNs in distinguishing hyperplastic from adenomatous polyps, thereby improving diagnostic specificity [63]. Transformer networks have been shown to reduce false alarms in AI-assisted colonoscopy by up to 30%, optimizing endoscopic workflow efficiency [67].

Recent advancements in AI for colonoscopy focus on hybrid architectures that integrate both CNN and transformer-based models, leveraging their complementary strengths. Hybrid models achieve high sensitivity without compromising specificity, making them particularly effective in high-precision medical imaging applications.

A hybrid AI system combining CNN-based segmentation with transformer-based classification demonstrated higher detection accuracy across multiple endoscopic datasets [68]. Studies indicate that hybrid models enhance polyp detection rates while minimizing false positives, striking a balance between sensitivity and specificity [68]. Clinical trials have confirmed that CNN-Transformer hybrids achieve robust generalization across diverse colonoscopy datasets, making them suitable for real-world applications [66].

The hybrid AI approach has shown the best balance of sensitivity, specificity, and false-positive reduction, making it an optimal choice for AI-assisted colonoscopy (Table 4). However, challenges such as computational complexity and real-time deployment feasibility remain areas for further research.

#### 3.7.3. Public Datasets for AI in Colonoscopy

The development of artificial intelligence (AI) models for colonoscopy relies on high-quality, diverse datasets to improve model generalization, reproducibility, and clinical applicability. Publicly available datasets play a crucial role in training, benchmarking, and validating AI-assisted colonoscopy models, ensuring robust performance across varied endoscopic imaging conditions. The following datasets (Table 5) are widely used in AI-driven colonoscopy research:

Publicly available datasets provide a standardized benchmark for AI model evaluation, fostering multidisciplinary research and collaboration across institutions.

Key benefits include improved model generalization, by training on diverse datasets, enhancing the robustness of AI algorithms across different patient populations and imaging conditions [78]. AI models can be compared objectively against standardized datasets, ensuring transparency and reproducibility, thus enhancing benchmarking and reproducibility [79]. These datasets include endoscopic images, video sequences, and clinical metadata, enabling the development of hybrid AI models integrating CNNs and transformers [80]. Despite the availability of public datasets, challenges remain in terms of dataset bias, limited diversity, and annotation inconsistencies.

#### 3.7.4. AI-Driven Quality Assessment Tools

AI-driven quality assessment tools monitor withdrawal time, mucosal exposure, and procedural completeness, addressing key factors influencing colonoscopy effectiveness.

AI models can assess optimal withdrawal speed to ensure complete mucosal inspection, improving polyp detection rates by up to 30% [81]. Imaging systems analyze real-time colonoscopic video streams to detect areas of suboptimal visualization and guide endoscopists in improving field-of-view coverage [82]. CNN-based quality control systems have demonstrated significant improvements in standardizing lesion detection, reducing operator-dependent variability [83].

Advanced AI algorithms are improving real-time lesion localization and motion tracking, addressing common challenges such as missed polyps and incomplete mucosal visualization. AI-powered tracking models reduce lesion miss rates by over 25%, particularly for small or sessile polyps [84]. Lesion localization accuracy is heightened by AI-enhanced localization tools that integrate computer vision and deep learning, improving accuracy in detecting and classifying polyps [85]. AI models assess colonic mucosal automated coverage mapping to guide the endoscopist towards unexamined regions, ensuring comprehensive inspection [86].

AI algorithms refine endoscopic images, improving contrast and resolution, leading to more accurate polyp identification [87]. Deep learning models dynamically adjust image parameters, enhancing visibility in low-light or shadowed mucosal areas [88]. Studies indicate that AI-powered enhancement techniques increase ADR by 20–35% compared to conventional colonoscopy [89].

## 4. Conclusions

The advancement of CRC diagnostics from traditional imaging techniques, such as barium enemas, to AI-assisted colonoscopy represents a transformative shift in precision medicine and early detection strategies.

Looking ahead, AI-driven multimodal integration, which combines colonoscopy imaging with biomarker analysis from liquid biopsy, holds promise for enhancing diagnostic precision and CRC surveillance. By merging real-time AI analytics, deep learning, and molecular diagnostics, this approach represents the next frontier in CRC screening and management, offering a more comprehensive, personalized, and efficient diagnostic pathway for patients.

Liquid biopsy, with its ability to detect ctDNA, CTC, and EVs, offers a minimally invasive alternative to traditional tissue biopsies. Its widespread implementation faces challenges due to high costs, variability in assay sensitivity, and the need for frequent testing to ensure dynamic tumor monitoring. These factors contribute to higher per-patient expenses, making liquid biopsy less cost-effective.

Conversely, AI-assisted colonoscopy emerges as a more cost-efficient and scalable solution. While the initial investment in AI-driven endoscopy systems requires substantial funding for hardware upgrades, software development, and clinician training, these costs can be rapidly amortized in a high-volume, multidisciplinary hospital setting. At an institutional level, AI-powered colonoscopy can optimize workflow efficiency, reduce missed lesions, and minimize the need for repeat procedures, ultimately leading to cost savings over time.

As these technologies continue to evolve, their integration into routine clinical practice has the potential to redefine the standard of care in CRC screening and surveillance, ultimately improving patient outcomes worldwide.

## Figures and Tables

**Figure 1 diagnostics-15-00974-f001:**
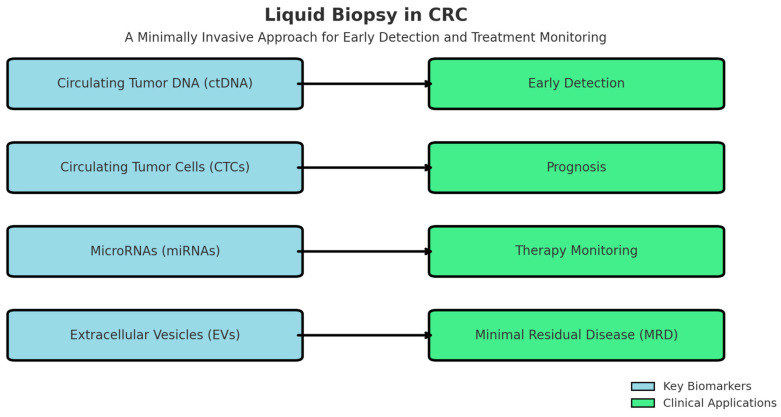
Clinical Application of Key Biomarkers.

**Table 1 diagnostics-15-00974-t001:** Gene function and mutation.

Gene	Purpose	Mutation & Role in Cancer	Associated Cancers
APC	Encodes a tumor suppressor protein involved in signaling, migration, and cell adhesion.	Mutations disrupt cell adhesion and proliferation control.	CRC
DCC	Produces netrin-1 protein, functions as a tumor suppressor.	Loss of function can lead to tumor development.	CRC, Esophageal Cancer
TP53	Tumor suppressor gene regulating cell-cycle arrest and apoptosis.	Mutations impair apoptosis, leading to uncontrolled growth.	CRC
BRAF	Regulates MAP kinase signaling, impacting cell differentiation and division.	Mutations can lead to hyperactive cell signaling.	CRC, Non-Hodgkin Lymphoma, Malignant Melanoma, Thyroid Cancer, Non-Small Cell Lung Cancer
PIK3CA	Encodes p110 alpha, a kinase in cell growth and survival pathways.	Mutations increase cell proliferation and survival.	CRC, Ovarian Cancer, Breast Cancer, Stomach Cancer, Lung Cancer, Brain Cancer
P53	Tumor suppressor controlling cell cycle, apoptosis, and DNA repair.	Mutations contribute to hereditary cancer risks.	Hereditary Cancers
KISS1	Suppressing metastasis formation by Kiss1R binding	Loss of function can enhance metastasis.	CRC
SMAD4	Regulates TGF-β pathway, controls DNA binding for tumor suppression.	Loss of function leads to enhanced tumor progression.	CRC, Polyposis Syndromes, Pancreatic Cancer
AKT1	Involved in oncogenesis, cell proliferation, survival, and angiogenesis processes.	Mutations drive uncontrolled cell growth.	CRC
K-RAS	Encodes a GTPase regulating cell division and apoptosis.	Mutations promote persistent cell signaling and growth.	CRC
CTNNB1	Critical for cell adhesion and epithelial layer formation.	Mutations contribute to abnormal cell growth.	CRC, Medulloblastoma, Ovarian Cancer

**Table 2 diagnostics-15-00974-t002:** Comparison between diagnostic methods.

Method	Mechanism	Sensibility	Specificity	Observation
Occult blood in feces (Guaiac Test)	Detects peroxidase activity in heme groups present in stool samples.	30–52% (increases to 90% with yearly use)	95.2%	Not specific to human hemoglobin, potential false positives from peroxidase-rich foods (e.g., raw vegetables, red meat). Patients should avoid NSAIDs 7 days prior to testing.
Feces Immuno- histochemical	Uses monoclonal or polyclonal antibodies to detect human hemoglobin in stool.	76.5%	95.3%	Exclusively reacts with human hemoglobin, thus being more specific than Guaiac Test. No dietary restrictions required. Recommended for population-wide screening.
DNA Analysis in fecal residues	PCR analysis detecting mutations in KRAS, APC, TP53 and elevated PDX1 levels.	52%	94.4%	Used to identify genetic markers associated with CRC risk.
Digital Rectal Examination	Initial evaluation method for symptomatic patients.	4.9%	97.1%	Not a screening tool, but useful for detecting rectal masses.
Flexible Sigmoidoscopy	Uses an endoscope to inspect the rectum, sigmoid colon, and descending colon (up to 60 cm).	58–75% (small lesions); 72–86% (advanced lesions)	94%	Reduces CRC mortality; should be performed every 5 years.
Colonoscopy	Direct visualization of the colon for polyp and abnormal tissue detection.	91%	94%	Gold-standard screening tool. Risks: Perforation (2%), hemorrhage (0.5% post-polypectomy), cardiovascular complications (arrhythmia, hypotension).
Endoscopic capsule	Swallowed camera capsule captures images of the digestive tract.	77%	59%	Primarily used for small intestine evaluation, limited for colorectal cancer detection.
Barium enema	Barium and air introduced into the colon to create contrast-enhanced X-ray images.	61–100%	100%	Alternative for patients who cannot undergo colonoscopy. Risks: Perforation (1 in 25,000 cases), mortality (1 in 55,000).
CT scan (Virtual Colonoscopy)	Advanced imaging using contrast-enhanced CT scans for colon evaluation.	Varies	Varies	Recommended for patients unable to undergo colonoscopy (e.g., anticoagulant users, those with pulmonary fibrosis, or sedative allergies).
Magnetic Resonance Imaging (MRI)	Creates detailed images to assess tumor characteristics and metastases.	75–90%	96%	Non-invasive, no ionizing radiation, superior for soft tissue resolution and staging.
Endorectal Ultrasound (EUS)	Uses high-frequency ultrasound with a saline balloon for 360° imaging of rectal walls.	69–97%	Varies	Key method for rectal cancer staging and local recurrence assessment
Positron Emission Tomography (PET Scan)	Assesses tumor staging, lymph node involvement, and distant metastases.	Varies	Varies	Used for comprehensive CRC staging, especially extrahepatic metastases.

**Table 3 diagnostics-15-00974-t003:** Key Biomarkers of CRC.

Circulating Tumor DNA (ctDNA)	Circulating Tumor Cells (CTCs)	MicroRNAs (miRNAs)	Extracellular Vehicles (EVs)
ctDNA has gained prominence as a highly specific biomarker for CRC detection. It allows for mutation profiling, treatment response assessment, and minimal residual disease (MRD) detection.	CTCs are cells shed by the primary tumor into the bloodstream, providing real-time insights into metastasis and treatment response.	miRNAs such as hsa-miR-21-5p and miR-221-3p have been identified as potential diagnostic biomarkers in CRC [51].	EVs secreted by tumor cells carry oncogenic cargo, including proteins, DNA, and RNA, which can serve as diagnostic tools for early-stage CRC detection [52].
Recent studies suggest that ctDNA analysis can detect KRAS, BRAF, and TP53 mutations, which play a critical role in CRC progression [53].	Studies suggest that a higher CTC count correlates with poorer survival outcomes in metastatic CRC [54].		
ctDNA-guided surveillance has shown superior sensitivity compared to carcinoembryonic antigen (CEA) levels in predicting disease recurrence [55].	Advances in single-cell sequencing of CTCs enable better molecular characterization of CRC subtypes, improving personalized treatment strategies.		

**Table 4 diagnostics-15-00974-t004:** Comparative advantages of CNNs, transformers, and hybrid AI models across multiple evaluation metrics [61,63,65].

AI Model Type	Accuracy (%)	False Positive Rate	Sensitivity (%)	Specificity (%)	Notable Applications
CNN-Based Models	~90%	Moderate	High (85–92%)	Moderate (80–88%)	Polyp detection, segmentation
Transformer Models	85–93%	Low	Moderate (80–88%)	High (90–95%)	Real-time video analysis, polyp localization
Hybrid AI Models (CNN + Transformer)	92–96%	Lowest	Very High (90–97%)	High (92–98%)	Combined polyp detection and classification

**Table 5 diagnostics-15-00974-t005:** Overview of existing open academic GI datasets.

Dataset	Findings	Size	Reference
CVC-ClinicDB (also named CVC-612)	Polyps	612 images	[69]
Endoscopy Artifact detection 2019	Endoscopic Artifacts	5138 images	[70]
ETIS-Larib Polyp DB	Polyps	196 images	[71]
KID	Angiectasia, bleeding, inflammations, polyps	2371 images and 47 videos	[72]
GASTROLAB	GI lesions	Some 100 s of images and few videos	[73]
El salvador atlas of gastrointestinal video endoscopy	GI lesions	5154 video clips	[74]
Kvasir	Polyps, esophagitis, ulcerative colitis, Z-line, pylorus, cecum, dyed polyp, dyed resection margins, stool	8000 images	[75]
Kvasir-SEG	Polyps	1000 images	[76]
Nerthus	Stool—categorization of bowel cleanliness	21 videos	[77]

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
