# Peer review of "Advancing Colorectal Cancer Diagnostics from Barium Enema to AI-Assisted Colonoscopy"

_diagnostics, 2025, doi:10.3390/diagnostics15080974_

Round 1
Reviewer 1 Report
Comments and Suggestions for Authors
Thanks the authors for the comprehensive review. I have several comments.
1) Table 2, please use the column name Sensitivity rather than Sensibility.
2) I would like to see a table summary in the Section 3.6 about different AI approaches. For example, what methods they are using, what model architectures they are using, etc.
3) Please add more AI-related discussions and references under Section 3.6 as there are way more studies talking about AI-assisted CRC diagnosis – different model architecture, different comparisons with classical clinical measurements, etc.
Author Response
Esteemed Reviewer 1,
Herein we, the Authors, will provide our replies to the commentaries provided in the first round of review for our manuscript. We hope that our response is consistent with the Reviewer’s expectations and that it answers the concerns raised prior.
For ease of reference, we will be including the Reviewer’s notes using quotation marks and italics with our response detailed below the quote.
Respectfully,
The Authors
“Thanks the authors for the comprehensive review.”
We thank the reviewer for the praise!
“1) Table 2, please use the column name Sensitivity rather than Sensibility.”
We thank the reviewer for the suggestion. We have changed the column name.
“2) I would like to see a table summary in the Section 3.6 about different AI approaches. For example, what methods they are using, what model architectures they are using, etc.”
“3) Please add more AI-related discussions and references under Section 3.6 as there are way more studies talking about AI-assisted CRC diagnosis – different model architecture, different comparisons with classical clinical measurements, etc”
We thank the reviewer for the suggestions. We have rewritten the AI chapter to be more through; to include more relevant references and have included tables to better emphasize the main points.

Reviewer 2 Report
Comments and Suggestions for Authors
A very brief but informative article. It starts from routine staging to gene expressions, liquid biopsy and in the end AI in imaging and diagnosis. The liquid biopsy part was very good , I suggest you have a graphical abstract for this part to summarise it.
The AI part didn't to be consistent, with different parts here and there. I suggest you rewrite this part. Also there are various fields for AI in CRC imaging( such as LN detection, complete response Prediction rating and...)
I think this part that is the main part of your article should be more inclusive.
Author Response
Esteemed Reviewer 2,
Herein we, the Authors, will provide our replies to the commentaries provided in the first round of review for our manuscript. We hope that our response is consistent with the Reviewer’s expectations and that it answers the concerns raised prior.
For ease of reference, we will be including the Reviewer’s notes using quotation marks and italics with our response detailed below the quote.
Respectfully,
The Authors
“A very brief but informative article. It starts from routine staging to gene expressions, liquid biopsy and in the end AI in imaging and diagnosis.
We thank the reviewer for the praise!
“The liquid biopsy part was very good , I suggest you have a graphical abstract for this part to summarise it.”
We thank the reviewer for the praise and for the suggestion. We have added a graphical representation to highlight the importante ond usage of Liquid biopsy.
“2) The AI part didn't to be consistent, with different parts here and there. I suggest you rewrite this part. Also there are various fields for AI in CRC imaging( such as LN detection, complete response Prediction rating and...)
I think this part that is the main part of your article should be more inclusive.”
We thank the reviewer for the suggestions. We have rewritten the AI chapter to be more through; to include more relevant references and have included tables to better emphasize the main points.

Reviewer 3 Report
Comments and Suggestions for Authors
Colorectal Cancer (CRC) is a rapidly escalating public health concern, highlighting the importance of early detection and the need to refine current screening methods. The development of Artificial Intelligence (AI) algorithms has significantly impacted the medical field, achieving remarkable success. The widespread application of AI technology in diagnosing and treating various types of cancer, particularly CRC, is now garnering substantial attention. In this paper, the authors review the evolution of CRC diagnostic tools, transitioning from conventional imaging methods to advanced AI-driven approaches.
Comments:
- The authors referenced 70 papers for this review article. However, many other relevant publications could have been included to provide a more comprehensive perspective.
- They presented a summary of previous research work, effectively consolidating existing knowledge in the field.
- While the review is thorough, it does not propose any new research ideas or innovative directions based on the analysis of previous publications.
The English could be improved to more clearly express the research.
Author Response
Esteemed Reviewer 3,
Herein we, the Authors, will provide our replies to the commentaries provided in the first round of review for our manuscript. We hope that our response is consistent with the Reviewer’s expectations and that it answers the concerns raised prior.
For ease of reference, we will be including the Reviewer’s notes using quotation marks and italics with our response detailed below the quote.
Respectfully,
The Authors
“The authors referenced 70 papers for this review article. However, many other relevant publications could have been included to provide a more comprehensive perspective.”
We thank the reviewer for the suggestion. We have cited more publication, that we belive are of relevance to the article, in our revised manuscript.
“They presented a summary of previous research work, effectively consolidating existing knowledge in the field.”
We thank the reviewer for the praise.
“While the review is thorough, it does not propose any new research ideas or innovative directions based on the analysis of previous publications.”
We thank the reviewer for the suggestions. We have rewritten the AI chapter to be more through; to include more relevant references and have included tables to better emphasize the main points. We hope this will better offer a new perspective.

Reviewer 4 Report
Comments and Suggestions for Authors
This study examines the evolution of colorectal cancer diagnostic methods and evaluates how AI-assisted colonoscopy and liquid biopsy have transformed diagnostic accuracy, clinical applicability, and cost-effectiveness. Its contribution to the literature is to present a comparative analysis with traditional methods, revealing how future diagnostic approaches can be improved with hybrid models.
Some corrections are listed below:
-Studies on AI-assisted colonoscopy and liquid biopsy were reviewed, but public datasets in this field were not included. Presenting public datasets in a table will facilitate multidisciplinary studies.
-It is stated that a systematic literature review was conducted in the study, but the inclusion and exclusion criteria, such as which years of studies were included and which studies were not, were not clearly presented. This process should be made more transparent by adding a visual such as a PRISMA diagram.
-It would be useful to add a table showing cost comparisons for traditional colonoscopy, AI-assisted colonoscopy, and liquid biopsy.
-A more detailed comparison of the advantages and disadvantages of different AI approaches should be made in the study. For example, it would be useful to add comparative performance results of CNN, Transformer or hybrid models.
Author Response
Esteemed Reviewer 4,
Herein we, the Authors, will provide our replies to the commentaries provided in the first round of review for our manuscript. We hope that our response is consistent with the Reviewer’s expectations and that it answers the concerns raised prior.
For ease of reference, we will be including the Reviewer’s notes using quotation marks and italics with our response detailed below the quote.
Respectfully,
The Authors
“-Studies on AI-assisted colonoscopy and liquid biopsy were reviewed, but public datasets in this field were not included. Presenting public datasets in a table will facilitate multidisciplinary studies.”
We thank the reviewer for the suggestion. We have included a table listing the main open source colonoscopy image databases.
-It is stated that a systematic literature review was conducted in the study, but the inclusion and exclusion criteria, such as which years of studies were included and which studies were not, were not clearly presented. This process should be made more transparent by adding a visual such as a PRISMA diagram.
We thank the reviewer for the suggestion. We have expanded the material and methods chapter to better explain our process.
-It would be useful to add a table showing cost comparisons for traditional colonoscopy, AI-assisted colonoscopy, and liquid biopsy.
-A more detailed comparison of the advantages and disadvantages of different AI approaches should be made in the study. For example, it would be useful to add comparative performance results of CNN, Transformer or hybrid models.
We thank the reviewer for the suggestion. We have rewritten the AI chapter to be more through; to include more relevant references and have included tables to better emphasize the main points.

Round 2
Reviewer 3 Report
Comments and Suggestions for Authors
Suggestions mentioned in the previous review
- The authors referenced 70 papers for this review article. However, many other relevant publications could have been included to provide a more comprehensive perspective.
- They presented a summary of previous research work, effectively consolidating existing knowledge in the field.
- While the review is thorough, it does not propose any new research ideas or innovative directions based on the analysis of previous publications.
In the revised manuscript, the authors addressed the first two suggestions from the previous review. However, the third suggestion remains unaddressed.
Comments on the Quality of English LanguageThe English could be improved to more clearly express the research.